

# CAMS-REG-v4: a state-of-the-art high-resolution European emission inventory for air quality modelling

Jeroen Kuenen[1], Stijn Dellaert[1], Antoon Visschedijk[1], Jukka-Pekka Jalkanen[2], Ingrid Super[1], Hugo Denier van der Gon[1]

[1]TNO, Department of Climate, Air and Sustainability, Princetonlaan 6, 3584CB Utrecht, The Netherlands
[2]FMI, Department of Atmospheric Composition Research, P.O. Box 503, FI-00101 Helsinki, Finland

*Correspondence to*: Jeroen Kuenen (Jeroen.Kuenen@tno.nl)

## Abstract

This paper presents a state-of-the-art anthropogenic emission inventory developed for the European domain for a 18-year time series (2000-2017) at a 0.1°x0.05° grid, specifically designed to support air quality modelling. The main air pollutants are included: NOx, $SO_2$, NMVOC, $NH_3$, CO, $PM_{10}$ and $PM_{2.5}$ and also $CH_4$. To stay as close as possible to the emissions as officially reported and used in policy assessment, the inventory uses where possible the officially reported emission data by European countries to the UN Framework Convention on
Climate Change and the Convention on Long-Range Transboundary Air Pollution as the basis. Where deemed necessary because of errors, incompleteness of inconsistencies, these are replaced with or complemented by other emission data, most notably the estimates included in the Greenhouse gas Air pollution Interaction and Synergies (GAINS) model. Emissions are collected at the high sectoral level, distinguishing around 250 different sector-fuel combinations, whereafter a consistent spatial distribution is applied for Europe. A specific proxy is selected for
each of the sector-fuel combinations, pollutants and years. Point source emissions are largely based on reported facility level emissions, complemented by other sources of point source data for power plants. For specific sources, the resulting emission data were replaced with other datasets. Emissions from shipping (both inland and at sea) are based on the results from the a separate shipping emission model where emissions are based on actual ship movement data, and agricultural waste burning emissions are based on satellite observations. The resulting
spatially distributed emissions are evaluated against earlier versions of the dataset as well as to alternative emission estimates, which reveals specific discrepancies in some cases. Along with the resulting annual emission maps, profiles for splitting PM and NMVOC into individual component are provided, as well as information on the height profile by sector and temporal disaggregation down to hourly level to support modelling activities. Annual grid maps are available in csv and NetCDF format (Kuenen et al., 2021).



# 1   Introduction

Emission inventories are the key starting point for understanding the causes and possible mitigation of air pollution. They provide information about the sources of air pollution, which can be used in environmental assessment models and air quality models to obtain information on levels of air pollution. In particular,
atmospheric dispersion models are using the information on the releases of air pollutants into the atmosphere to calculate levels of air pollution in various geographical domains to get a better understanding of the relation between emission sources and air pollutant concentrations (Belis et al., 2020; Liang et al., 2018). When inventories span a longer period the trend in air pollution and exposure can also be assessed, for example to identify the impact of changes in industrial processes,  fuel mix or implementation of policies (Buonocore et al.,
40  2021).

At the same time, emission inventories are the backbone of policies that control air pollution and climate change. In the climate change community, the United Nations Framework Convention on Climate Change (UNFCCC) relies on emission inventories to provide information on the reduction of emissions and progress towards future reduction commitments related to the Kyoto Protocol and the Paris Agreement. In the air pollution community,
the UNECE Convention on Long-Range Transboundary Air Pollution (CLRTAP) (UNECE, 2012) and the EU National Emission Ceilings Directive (NECD) (European Commission, 2016) set reduction commitments for air pollutant emissions. In this framework,  emission inventories are used to identify mitigation options and to check progress towards objectives. In view of these policy needs, each country that is part of the Convention is required to develop an emission inventory that meets the requirements set for these inventories annually. These
requirements have been standardized over the last decades by prescribing methodologies for both greenhouse gases and air pollutants. These methodologies, documented in the IPCC Guidelines for National Greenhouse Gas Inventories (Eggleston et al., 2006) for greenhouse gases and the EMEP/EEA Air Pollutant Emission Inventory Guidebook (EEA, 2019a) for air pollutants, provide a set of default methodologies that each country shall use to establish the emission inventory for each source. However, they explicitly say that if a country has better
information than the default methodologies provided by the guidance, it should be used.

Apart from the national total emissions by sector, both CLRTAP and NECD require countries to submit gridded emission inventories to support air quality modelling and assessment at (sub-)national and European level. While until 2016, the requirement under CLRTAP was to report at a horizontal resolution of 50x50km2, the resolution for both CLRTAP and NECD has been increased to 0.1°x0.1°, which is equivalent to roughly 5-10km.. However,
most of the EU Member States do report gridded emission data that are good quality, some of them and most of the non-EU countries submit incomplete or erroneous data or no data at all (EMEP, 2017). Therefore, the merged submitted gridded inventory data  does not yet provide a complete and reliable inventory for the entire European domain that is usable for air quality modelling.

Next to the official inventories various initiatives have been providing gridded emission inventories as this
information is a prerequisite for global, national and local air quality studies and policies.   Globally, the EDGAR emission inventory was developed since the 1990s  as a bottom-up emission inventory using activity data (e.g. energy statistics) and emission factors (Crippa et al., 2018). Other important global inventories include ECLIPSE (Klimont et al., 2017), CEDS (McDuffie et al., 2020) and CAMS-GLOB-ANT (Doumbia et al., 2021). At the European level, the TNO_MACC inventories have been developed in the MACC (Monitoring Atmospheric





Composition and Climate) FP7 project since 2007 (Kuenen et al., 2014). The MACC project has evolved into the Copernicus Atmosphere Monitoring Service (CAMS) under the umbrella of the EU Copernicus programme (https://www.copernicus.eu/en). CAMS identified a continuous need for up-to-date emission information that can be used in support of air quality production and forecasting systems at both the global and the European scale. To fulfill this need at the European scale  the CAMS regional inventory (CAMS-REG) was developed. This paper

describes the methodology used to derive the CAMS-REG inventory, in particular its version 4 which covers the years 2000-2017. Apart from the use in CAMS, the data are freely available for air quality modellers and other scientists who are in need of emission information at the European scale. To make the inventory fit for purpose CAMS-REG not only provides the gridded emissions but also default profiles for typical emission height by source type, temporal profiles to distribute emissions over the year, and  chemical speciation of PM and NMVOC

emissions.

## 2    Methodology

The CAMS-REG emission inventory focuses on the main air pollutants ($NO_x$, $SO_2$, NMVOC, NH3, CO, $PM_{10}$ and $PM_{2.5}$). $CH_4$ is also included because of its role in atmospheric chemistry. For this inventory, the general methodology is illustrated in Figure 1. The strategy was to use official reported emission data from national

inventories for both the greenhouse gases and the air pollutants where possible, recognizing that these often contain specific national information which increases the accuracy of the data compared to estimates at continental or global scale (see Sect. 2.2.2 and 2.2.3). At the same time, it is clear that in specific cases these data have shortcomings e.g. in the form of missing or inaccurate data (EMEP, 2020). Therefore, an alternative emission dataset available from the IIASA GAINS model (IIASA, 2018) was used to fill in the gaps or replace

any data which are considered of insufficient quality (see Sect. 2.2.1 and 2.2.4). All of this was done at the level of annual sectoral emissions by country (not distributed in space). As a next step, the dataset holding emissions from these different sources was spatially distributed in a consistent manner using relevant proxies for each source (see Sect. 2.3). In addition, all shipping emissions were excluded and taken from a different data source (see Sect. 2.4), to allow a consistent approach towards all shipping emissions given the mix of national and

international shipping. For agricultural waste burning a similar approach was followed (see Sect. 2.5) given the limited reporting of emissions from this source.

This methodology was applied for UNECE-Europe, which refers to all European countries including Turkey (as a whole) and Russia (only the European part, until 60°E) at the eastern side. The domain stretches between 30°W and 60°E, and between 30°N and 72°N, including all the European countries fully within the domain. Emissions

from countries outside of Europe but still part of the rectangular domain (most notably North Africa and the Middle-East) were taken from EDGAR-v4.3.2 (Crippa et al., 2018) to complete the overview of anthropogenic emissions for the entire domain. Finally, only those sources contributing to the national total emissions in the inventory reporting system were included, all (semi-)natural sources were excluded for as far as reported, which is in line with the sources included in official national total emissions used for compliance assessment.




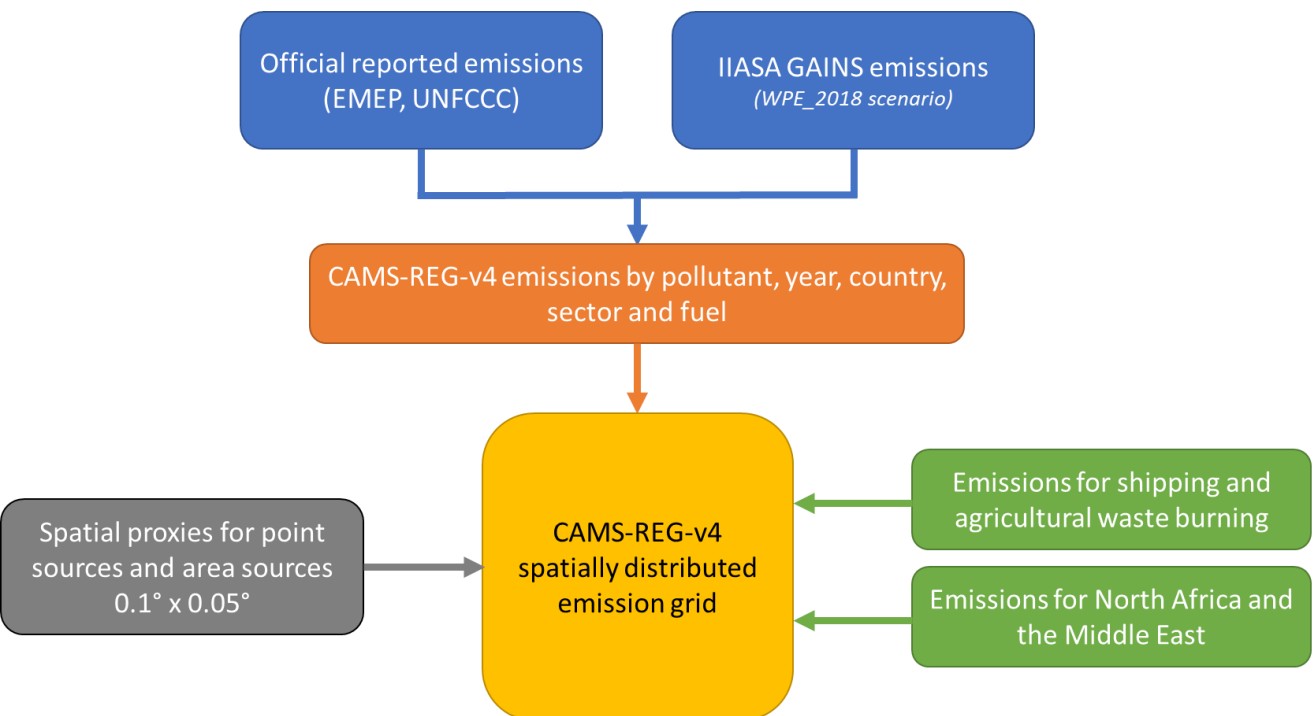

**Figure 1: General methodology applied for the CAMS-REG emission inventory.**

## 2.1 Source sector definition

The reported data from EMEP and UNFCCC (for $CH_4$) as well as the IIASA GAINS emissions were converted to a newly defined sector format, which combined the level of detail available in various datasets. A full overview of the sector definitions is provided in the supplementary information (SI, Table S1). This new sector and fuel definition was developed primarily since none of the existing classifications did contain all the necessary details. Also it gave the opportunity to introduce a hierarchical and numerical structure which allows simple aggregation and disaggregation of emission sectors where necessary. The sectoral structure defines 209 individual source categories at the highest level of detail, taking into account the highest detail in sectoral emissions for each of the data sources. Each sector can be aggregated up to a minimum of 7 main groups: energy industry, manufacturing industry and product use, road transport, non-road transport, small combustion activities, agriculture and waste. In practice, the emission data were processed at the highest sectoral detail possible, restricted by the level of detail in the different data sources (EMEP, UNFCCC and IIASA GAINS).

Apart from the detailed source sector definition, also an aggregated sector level was defined which is used as the default aggregation level in which the gridded emissions are provided. This aggregation is based on the GNFR sector level, which is an aggregation of the NFR (Nomenclature For Reporting) that is used as the basis for reporting spatially distributed emissions of air pollutants by European countries. A complete overview of the GNFR sectors in this inventory is given in the supplementary information (SI, Table S1).



## 2.2 Emission data collection and processing

### 2.2.1 GAINS emissions

The GAINS model is developed by IIASA to explore emission control strategies for air pollutants and greenhouse gases by modelling the impact of possible measures. Originally developed for Europe, it now covers most regions of the world. The model is used to underpin policies such as the UNECE Gothenburg Protocol (UNECE, 2012) and the EU National Emission Ceilings Directive (European Commission, 2016). For this inventory, the most recent emission data from the model was taken, as it was incorporated in the

CEP_post2014_CLE scenario updated in December 2018 (IIASA, 2018). This scenario takes into account historical emission data up to 2015, and future emissions for 5-yearly intervals (2020, 2025, 2030) were modelled based on the latest available information on activity data and control measures available at the time. The emission data were obtained at the level of detailed source categories and fuels for each country for 5-yearly intervals (2005, 2010, 2015, 2020). Linear interpolation was used to estimate emissions for each of the years in between.

To estimate emissions prior to 2005, an earlier GAINS dataset which was used in the TNO_MACC-II inventory (Kuenen et al., 2014) and does include the year 2000 was used to extrapolate the trend backwards in time until 2000.

The GAINS sector and fuel classifications were converted to our own sector and fuel definitions (see SI: Table S1 for definition, Table S4 for the links). For industrial combustion, an additional step was needed since the

GAINS sector classification has most industrial combustion aggregated to one industrial combustion sector. To split these over various industrial sectors, a specific bottom-up emission inventory was set up for the industrial sectors. This bottom-up inventory uses energy consumption from the IEA energy statistics combined with default emission factors to calculate emissions per pollutant, country, year and industrial sector. The share of each sector in this bottom-up inventory was then used to disaggregate the GAINS industrial combustion emissions over the

different industrial sectors as identified in this inventory (see SI, Table S1).

### 2.2.2 Reported emissions for greenhouse gases

$CH_4$ emissions for 2000-2017 (based on reporting year 2019) were obtained from the national inventory submissions to the UNFCCC (CRF tables) (UNFCCC, 2019). Emission data at CRF level were extracted from the CRF tables and combined into a single database. For categories 1A1-1A4 which concern emissions from the

combustion of fuels, emission data were collected for each individual fuel. Subsequently, the CRF sectors were converted to our own sector definitions (see SI, Table S3).

### 2.2.3 Reported emissions for air pollutants

Official reported emissions (reporting year 2019) were obtained from CEIP for the years 2000-2017 (CEIP, 2019), containing sectoral emissions reported under the EMEP reporting requirements, for all the countries for

which data were available and for all air pollutants included in the scope of this inventory. Reporting follows the NFR (Nomenclature for Reporting) structure, which was converted to our own sector definitions as developed for



this inventory (see SI, Table S2). Whereas the UNFCCC emissions from combustion activities and IIASA GAINS emissions are available per main fuel type, the EMEP emissions are not. Therefore, the relative distribution of fuels for each pollutant, year, country and sector from the GAINS emissions dataset was used to
add the fuel split to the dataset. Where for a specific combination, no emission was available from the GAINS dataset for the same pollutant, year, sector and country, an average fuel split was used which was calculated by taking the GAINS data for the sum of all years 2000-2017. Ultimately, if also this average split was not available, default fuel splits were calculated for each pollutant and sector based on the total emissions from GAINS for the pollutant and sector (in all countries and all years).

**2.2.4   Combination and processing**

As a next step, the reported data were quality checked to decide in which cases these are fit for use. In the quality checks, it was taken into account that the greenhouse gas emissions from Annex I Parties have been reviewed on an annual basis for many years. For air pollutants, this annual review cycle is in place since 2017 as part of the NEC Directive, thus covering only the EU Member States. While this annual review cycle has improved
reporting by countries over time, still shortcomings are identified for EU Member States (IIASA, 2019). For non-EU countries, the completeness and quality of the inventory data differs significantly between countries, and for some countries no national emission inventory is available at all (EMEP, 2020). All in all, a thorough check on completeness and accuracy is key before using reported emission inventory data for air quality assessment. Therefore, as a starting point the reported data from each EU Member State (including the United Kingdom and
also including Iceland, Norway and Switzerland) were used, whereas reported emission data from other countries were not used. Hereafter these 31 countries are referred to as the EU+ countries. While for all of these other countries (consisting of Turkey, Balkan countries, and former Soviet Union countries) in some cases reported emission data are submitted, no consistent emission inventory for air pollutants is available on an annual basis. Therefore GAINS emissions were used for these countries. And while for some of these countries GHG
emissions are being reported to UNFCCC, for consistency reasons GAINS data were also used for the $CH_4$. Figure 2 shows the data sources for each European country.



**Figure 2 Emission domain and choice of data source for each country. Green: reported data, orange: reported data with significant corrections/gapfilling, red: GAINS emissions, blue: countries outside of UNECE-Europe, gridded emissions from EDGAR-v4.3.2 inserted for these locations**

Next, a quality check is performed for the 31 countries for which reported data were used. These checks focussed on completeness (emissions reported for each GNFR sector where emissions are expected to occur), time series consistency (trend in reported emissions as expected, no missing years), and checks on the distribution of emissions over sectors to identify possible missing sectors. Compared to the assessment of reported data for the earlier TNO_MACC-II inventory (Kuenen et al., 2014), the number of gaps and inconsistencies was significantly smaller, and no major issues of such nature were identified. For 3 countries (Romania, Malta and Lithuania), emissions prior to 2005 were found to be incomplete and/or inconsistent with later years. This is likely related to the fact that under the NEC Directive, 2005 is the base year and there is relatively little attention for earlier years. For these 3 countries, the 2000-2004 emissions were replaced with an extrapolation of the 2005 emissions based on the trend in GAINS emissions, per GNFR category.

At the same time, larger inconstancies were identified for agricultural emissions. For NMVOC from animal husbandry and manure application, a methodology to estimate emissions was only recently included in the





EMEP/EEA Guidebook (EEA, 2019a), which led to inconsistent and incomplete reporting by countries. Therefore it was chosen to leave this source out of the CAMS-REG-v4 inventory. For $NO_x$ from agriculture, also reporting is found to be inconsistent between countries. In addition, one of the main sources of agricultural $NO_x$ emissions is soil $NO_x$, and many air quality models have separate modules to calculate soil $NO_x$ internally. To avoid double counting it was therefore decided to exclude $NO_x$ emissions from agriculture.

Minor issues that were found include mainly gaps or outliers in a specific year for a specific sectors. These were identified by looking at year-to-year changes for each country and pollutant at the level of GNFR categories (aggregated sectors) and take those cases where the average importance of the GNFR category in the national total was above 3%, and at the same time there was a change of more than 5% from one year to the next in the time series. Where this was based on reported data, the underlying detailed sector data were checked, and gaps or
other errors were identified these were fixed using interpolation and/or extrapolation or by keeping emissions constant from the previous/next year (based on a manual case-by-case assessment).

Finally, some other modifications were made to the dataset to make it consistent and fit for purpose for the spatial distribution:

- Emissions of NMVOC from natural gas production and distribution systems (NFR category 1B2b) were split into production, high-pressure distribution and low-pressure distribution based on the relative contribution of these subsectors in GAINS to total emissions of $CH_4$.
- Emissions from combustion in energy industries (excl. power/heat plants and refineries) were split into fuel consumption in coal mines, oil extraction, gas extraction and coke ovens (see SI Chapter 1 for details).
- Emissions from road transport are available at different levels of aggregation (different vehicle type groups) which were harmonized. Also, a road type split between highway and non-highway (urban and rural) emissions was added based on information obtained from the COPERT model (Ntziachristos et al., 2009). More details are provided in the SI (Chapter 3).

**2.3 Spatial distribution**

Each combination of sector and fuel was assigned a specific proxy for the spatial distribution. The proxy is a variable which is available in gridded form (at the resolution of 0.1° x 0.05°) and can be used to mimic the spatial distribution of the emission source. The proxy is defined as the fraction of the national total to be allocated to this grid cell, and the sum of fractions for each country always equals one.

Some examples of proxies include the network of highways in each country with traffic intensities, which is used to distribute emission from road transport on highways, and a list of emissions from individual power plants and their emission strength for specific pollutants is used to distribute emission from power plants. The summary of proxies used for each sector is provided in Table 1, and a complete overview of the selected proxies for each source is provided in the Supplementary Information (SI, Table S5).






**Table 1 Summary table with main proxies per GNFR source category**

| GNFR category | Main proxies used |
|---|---|
| A Power plants | • E-PRTR & LCP reporting combined with Platts-WEPP<br>• CORINE land cover 2012 Industrial area |
| B Industrial sources | • E-PRTR<br>• Own point source database<br>• CORINE land cover 2012 Industrial area |
| C Other stationary combustion | • Population density<br>• CORINE land cover 2012 Arable land (for stationary agricultural emissions)<br>• Own wood consumption map |
| D Fugitives | • Own point source database<br>• Population density |
| E Solvents | • CORINE land cover 2012 Industrial area<br>• Population density |
| F Road Transport | • Road network<br>• Population density |
| H Aviation | • Airports |
| I Off-road | • Population density<br>• CORINE land cover 2012 Industrial area, Arable land |
| J Waste | • E-PRTR<br>• Own point source database<br>• Waste water treatment plants<br>• Population density |
| K Agriculture livestock | • FAO gridded livestock |
| L Agriculture other | • CAPRI model distributions<br>• CORINE land cover 2012 Arable land |

For a number of the sectors, point source information was used in the spatial distribution. This concerns power plants, industrial sources, airports and waste water treatment plants, these are described in Sect. 2.3.1 to 2.3.3.
Non-point source distribution proxies are described in the other subsections.



### 2.3.1 Public power and heat plants

For public power and heat plants, three different datasets were combined. E-PRTR (the European Pollutant Release and Transfer Register) collects facility level emission data for all EU Member States, with as main goal to inform the European citizens about pollutants transfers and releases in their area. The data are reported on an annual basis since 2007 (before that three-yearly since 2001) and publicly available (EEA, 2019b). In addition to E-PRTR, EU Member States are required to report information (fuel consumption and emissions) at the level of individual installations (stacks) of large combustion plants annually. This requirement follows from the LCP Directive (European Commission, 2001), superseded by the Industrial Emissions Directive (European Commission, 2010). Thirdly, a commercial dataset on power plants known as Platts-WEPP was used, which records characteristics of power plants worldwide, such as the technology type, fuel type, capacity, etc. (Platts, 2017).

In processing the E-PRTR and LCP reported emission data at facility level, a sector designator was used to select the facilities corresponding to the public power and heat sector. Matching and linking of identical facilities in the LCP and E-PRTR datasets was partly possible using a joint 'national ID' field. Linking was then manually completed based on similarities in facility name and location and each individual facility was assigned a new and unique identifier. Emissions of $NO_x$, $SO_2$ and dust reported to the LCP dataset, were supplemented with an estimate of $CO_2$ emissions based on reported fuel use and default $CO_2$ emission factors (Eggleston et al., 2006).

Emissions reported to E-PRTR and LCP datasets were then combined by facility, pollutant and year using the new joint ID field. Since the scope of the E-PRTR is the most complete[1], where available the E-PRTR emissions were selected. Where E-PRTR emissions were missing, but LCP emissions had been reported, LCP emissions were used instead. Since the scope of LCP and E-PRTR reporting are not identical, the LCP emission values used for gapfilling were adjusted based on the average ratio between the E-PRTR and LCP emission values for the years where both were reported. A final step of emissions gapfilling was performed using the ratio between reported emission values for $CO_2$ and the other pollutants. When both an air pollutant and $CO_2$ are reported together for several years, the average ratio between the emission values was multiplied with the reported (or calculated) $CO_2$ value for years the pollutant had not been included. To avoid introducing outlier emission values, the E-PRTR reporting threshold were applied as a maximum emission value in gapfilling.

The facility level emissions were then split by fuel type. This was done by calculating a proxy emission value using the LCP reported fuel input by fuel type and country-, fuel- and pollutant-specific emission factors from the IIASA GAINS model at the country, fuel and pollutant specific level. The relative contribution of each fuel type was then applied to the reported emission value to distribute it to the various fuel types. Where no fuel input data was available, the unit fuel type from the Platts WEPP dataset was used instead to assign the emission value to one or multiple fuel types. Finally, for the remaining facilities the fuel type used was searched for online to fill the gaps.

Several checks were then performed to compare the total emissions by country with the reported UNFCCC and EMEP sector totals for public power and heat production. This check led to the identification of unrealistically

---

[1] LCP reporting is only for facilities with a thermal capacity >50 MW thermal, and covers emissions of NOx, SO2 and dust





high emission values for some facilities, where there had evidently been an error in reporting. In these cases, the erroneous value was removed (and then gapfilled following the routine described above) or lowered in case of a likely unit error (e.g. factor 10 too high).

The final step was the assignment of point source and area source emissions to GNFR A. In case the reported sector level emissions were higher than the processed facility level emissions, the remainder is assigned as area source emissions under GNFR A, representing the smaller facilities which are below the threshold for reporting. The processed facility level emissions were then assigned as point source emissions under GNFR A. For some countries and years, the total facility level emissions were higher than the reported sector total. In that case the
emissions for all facilities were scaled down to arrive at the country sector total emission value. For the point source emissions, the facility coordinates included in the E-PRTR dataset were used for spatial distribution of the emissions. For some facilities, coordinates were added or corrected manually when they were found to be incorrect or missing. For the area source emissions, the CORINE dataset was used to spatially allocate emissions to areas with industrial activity (see Sect. 2.3.5).

**2.3.2    E-PRTR for industrial sources**

For industrial point sources, similar to the power plants the main sources of information for the EU+ was E-PRTR, combined with various online sources and one commercial industrial directory, and the TNO point source database for the other countries in Europe (see Sect. 2.3.3). For industrial emissions in particular the E-PRTR registry is the most complete and best available database for the EU(28)+ but it nonetheless frequently contains
facilities that have an incorrect sector code or erroneous or missing emission data. Compared to power plants, industrial point sources in E-PRTR are much more numerous and diverse in type, and as a consequence a somewhat less detailed approach had to be followed to correct any pressing deficiencies in E-PRTR.

For oil refineries and integrated iron and steel plants, external lists of all existing plants in the EU+, including operational status, were consulted to extract all corresponding records from E-PRTR (years 2001, 2004 and
annually from 2007 onwards) regardless of the E-PRTR sector code. Any missing facilities were added. Complete lists of existing/operational oil refineries in the EU+ were available from for instance the OGJ Worldwide Refining Survey, or online directories such as the Wiki site "A barrel full" (A barrel full, n.d.). For integrated iron and steel plants, the extensive commercial Plantfacts Capacity Database was consulted to extract any facilities operational after the year 2000 from E-PRTR and to complete any missing plants. Plantfacts data is
an online source (World Steel Dynamics, n.d.) which was initially compiled and maintained by the German VDEH Steel Institute.

Next, based on E-PRTR sector classification, the following types industrial facilities were selected from E-PRTR (including EPER data for 2001 and 2004):

- Coal mines
- Coke ovens not belonging to iron and steel plants
- Secondary iron and steel smelters and foundries
- Chemical plants
- Non-ferrous metal plants (primary and secondary, incl. aluminium)



- Non-metallic mineral plants (e.g. cement, lime, glass etc.)
- Paper and pulp plants
- Waste incinerators without energy recovery
- Landfills without energy production
- Other waste disposal plants (such as composting plants)
- All other industrial facilities (except oil and gas production/transport/processing); grouped as "Other
industry"

All industrial facilities extracted from E-PRTR have been subjected to an brief individual check for plant
characteristics, to ensure that the plants were assigned the right industrial activity. This elaborate process used E-
PRTR plant names and locations to identify the true primary industrial activity of the E-PRTR facility through
online search, as the true main activity sometimes proved different from what E-PRTR indicated.

The basic aim for industrial point sources was to compile a complete emission time-series for plants that
appeared to have been operational in the period 2000-2017, but also to take plant closures, production stops and
emission data below the reporting thresholds into account. For oil refineries and integrated iron and steel plants
any missing (but expected) $CO_2$, $NO_x$ and $SO_2$ data were estimated, in addition to NMVOC emission data for
refineries specifically and CO and PM data for iron and steel plants. For all other industrial activities the basic
assumption used in gapfilling was that all operational plants must have emission data for each relevant pollutant,
for each reporting year. So principally all emission data missing in this sense have been gapfilled for each
facility, by assuming the average of the emissions reported by that facility for other years, unless:

- emission data reported for other years was ever close to the threshold (missing emission data is below
  threshold);
- emission data are missing at the beginning or end of a time series (facility may report missing emission
  data under a different facility ID in earlier or later years, or the plant was modernized or closed);
- a facility did not report any emission data at all for a specific year (facility is assumed to be temporarily
  shut down).

### 2.3.3    Other point source proxies

An **own point source database** was developed for the spatial distribution of point source emissions in earlier
European inventories (Denier Van Der Gon et al., 2010; Kuenen et al., 2014). Since this point source dataset is
only available for the year 2005, it is only used for specific sectors where no point source data could be extracted
from E-PRTR.

**Airports** have been included as point sources in this dataset. The contribution of each airport to the country total
for this sector was calculated based on flight statistics per airport (Eurostat, 2019). First, a split was made at
country level between passenger and freight traffic using the number of flights in the country as a whole.
Thereafter, freight traffic was distributed to individual airports using the tonnage of freight, and passenger traffic
was distributed using the number of passengers per airport. This was done on an annual basis, to allow for
changes in time (e.g. opening of a new airport in a different location). Eurostat data was only available from
2003, for 2000-2002 the distribution is based on the Eurostat statistics as of 2003.





For **domestic waste water treatment**, an EEA dataset on urban waste water treatment plants was used (EEA, 2014), which includes plant coordinates. This dataset provides the flow and capacity or urban waste water treatment plants in all European countries, from which their share in total emissions was inferred.

### 2.3.4 Population density

The LandScan Global dataset (Oak Ridge National Laboratory, 2017) for population density was obtained for the years 2005, 2010 and 2015 at high spatial resolution (~ 1km). A country mask was combined with this dataset to allocate each grid cell to a specific country. For cells that include a border between countries the relative area of each country was used to split up population for the two (or more) countries.

Based on the population of each grid cell, an additional qualification was made whether the cell was allocated as
urban or rural. The definition of urban and rural areas however differs significantly between different regions of the world and between countries. To ensure consistency across the domain, a fixed value of 250 inhabitants per $km^2$ was chosen. Above this value the cell classifies as urban, below it classifies as rural. This results in around 75% of the people living in urban areas, which corresponds well to the urban population percentage for the EU as published elsewhere (World Bank, 2018).

As a final step the data was converted to a resolution of 0.1° x 0.05° for the entire domain, separately for urban, rural and total population.

### 2.3.5 Land cover

The CORINE Land Cover dataset (Copernicus Land Monitoring Service, 2016) was obtained from Copernicus Land Monitoring. This dataset has a resolution of approximately 100 metres and for each grid cell the main use
type is given. These high resolution grid cells were aggregated to 0.1° x 0.05°, and for each of these larger grid cells the fraction of different main use types was calculated by adding up the amount of grid cells with this main use type and normalising the totals per 0.1° x 0.05° grid cell.

The dataset identifies around 45 different use classes. For this work, 3 different proxies were extracted from this dataset:

- Industrial area (taken as the sum of "Industrial or commercial units", "Port areas" and "Construction sites")
- Arable land (taken as the sum of "Non-irrigated arable land", "Permanently irrigated land", "Complex cultivation patterns" and "Land principally occupied by agriculture, with significant areas of natural vegetation")
- Rice fields

Similar to the population proxies, a country mask was added to the dataset to be able to calculate a distribution map for each country separately.



### 2.3.6 Road network

For road transportation, shape files from Open Transport Map (OpenTransportMap, 2017) and Open Street Map
(Open Street Map, 2017) were obtained for the entire European domain.

While Open Street Map contains the road network across Europe, OTM adds the traffic volumes to these and
divides the roads in different classes (main roads, and from first class to fifth class roads). The two datasets were
merged and the traffic volumes from OTM were used as a proxy for the emissions. However, especially for
smaller roads in many cases the traffic volume was not available and traffic volume had to be estimated for these
cases. To do this, first a relation between the traffic volume and the population density in each grid cell was
determined based on all the grid cells with a known traffic volume, per country and per road class. Then this
relation was used to estimate the traffic volume where this was not available, based on the population. Since
smaller roads are expected to contain mostly local traffic, this approach is expected to represent reality
reasonably well.

The traffic intensity map was classified per country, per vehicle type and per road type. The latter refers to the
separation between highway (assumed equivalent to main roads) and non-highway emissions. Thereafter, the
non-highway emissions were split between urban and rural by means of overlaying the traffic intensity map with
a population map (which includes rural and urban shares, see 2.3.4). In this way, 18 different traffic intensity
maps were created (6 vehicle types, 3 road types). As a final steps, this dataset was combined with a country
mask to introduce the country codes similar to the population and land use distributions, and the map was
subsequently normalized with the country total traffic intensities to create the final proxy map.

### 2.3.7 Other proxies

For agriculture, a number of specific agricultural proxies have been used:

- **Gridded livestock** of the world (FAO, 2010), available from the UN Food and Agricultural Organisation
(FAO) has been obtained per animal type and converted to a 0.1°x0.05° resolution.
- Distributions representative for **manure application and fertilizer application** have been extracted
    from the Common Agricultural Policy Regionalised Impact (CAPRI) modelling system (CAPRI, 2020).
    This model is a global partial equilibrium model for the agricultural sector with a focus on the European
    Union, developed in a series of EU research projects and is now used by the European Commission to
underpin agricultural policies.

Given the importance of **residential wood combustion** for some pollutants, a specific distribution proxy was
developed to represent wood combustion emissions, taking into account population density and also proximity to
wood. Using this proxy, most of the wood consumption is allocated to rural areas especially those near forested
areas. Despite this modification to the distribution of residential wood combustion, an overallocation of the
emissions in urbanized centres may still be present in the spatial distribution (Timmermans et al., 2013). In
practice it is observed that the distribution also depends on national circumstances, e.g. bans on using wood or
coal in urban areas, making it difficult to derive a generic distribution at European scale.





Finally, for **high distribution gas networks** and for **rail transport**, specific maps are available to spatially distribute emissions. These are similar to those used in the earlier TNO_MACC inventory (Kuenen et al., 2014).


## 2.4 Shipping emissions

Emissions from shipping can be divided into sea shipping and inland shipping, but also into domestic and international. The latter is typically used in official reporting, where domestic refers to a ship leaving and arriving in the same country, irrespective of the route. International shipping however is not a primary part of national
inventory reporting, and therefore reporting is more incomplete and inconsistent. Given the importance of shipping for emissions and air quality at European scale (Jonson et al., 2020; Viana et al., 2014), shipping emissions from national reporting are replaced with an alternative based on a consistent modelling approach. The STEAM model (Jalkanen et al., 2012; Johansson et al., 2017) provides global emissions at high resolution based on AIS (Automatic Identification System) records that track the whereabouts of ships around the world. The
STEAM model then computes emissions from shipping for CO2 as well as major air pollutants based on the generation of shipping routes from the AIS signals, and emission characteristics based on the characteristics of each ship. For the CAMS-REG inventory, the STEAM model was run at a resolution of 0.1°x0.05° for the European domain covered by this inventory, and for the relevant pollutants ($NO_x$, $SO_2$, CO, NMVOC, PM, CO2), where PM was speciated into EC, OC, $SO_4$ and ash. The main limitation for this inventory was that STEAM data
were only available from 2013 onwards, given the introduction of the AIS system around this time. Therefore, for all the years prior to 2013 the emission data have been extrapolated backward in time using a separate estimate of shipping emissions per year, pollutant and sea area using historic activity data and information on fuel quality regulations and policies. This extrapolation only concerned the total emissions per sea region, whereas the spatial distribution before 2013 is assumed constant. Therefore, the shipping emissions for the years 2013-2017 are more
certain than the pre-2013 data when no AIS data were of much lower quality and/or not available at all.

## 2.5 Agricultural waste burning

Field burning of agricultural waste is a separate reporting category in national inventories. Formally agricultural waste burning (AWB) is forbidden in the EU. It may however still happen albeit illegally, by accident or possibly
under certain exemptions. As a result the reporting of this category is highly variable between countries and inconsistent. In previous TNO-MACC (Kuenen et al., 2014) and CAMS-REG inventories this category was gap-filled using data from the IIASA-GAINS model (IIASA, 2018). With the on-going development of fire detection with global satellite products it was now possible to include an AWB emission estimate based on earth observation. For this we used the CAMS Global Fire Assimilation System (GFAS) which assimilates fire
radiative power (FRP) observations from satellite-based sensors to produce daily estimates of wildfire and biomass burning emissions (Kaiser et al., 2012). The GFAS data output includes spatially gridded Fire Radiative Power (FRP), dry matter burnt and biomass burning emissions for a large set of chemical, greenhouse gas and aerosol species. Data are available globally on a regular latitude-longitude grid with horizontal resolution of 0.1 degrees from 2003 to present. By overlapping these data with land use maps the emission from biomass burning





on agricultural fields was derived. To match with the annual total emission data and complete timeseries in CAMS-REG the available high resolution GFAS data over the years 2003-2018 were processed to derive at an annual average with a monthly distribution pattern and average spatial distribution map which can be applied to all years in the time series. It is important to note that in the case of Europe this approach only makes sense when data of a resolution of 0.1°x0.1° are available because in many European countries land use is mixed and forests 465 and agricultural lands may occur in the same pixel if the resolution is not high enough. In fact even at the resolution of 0.1°x0.1° this still introduces uncertainty. Nevertheless, it is by far the best resource available for this emission source. For further details we refer to the Chapter 2 in the SI.

## 2.6    Non-European countries

Apart from the European emissions, also emissions from just outside the European borders may influence air quality in the study area. Therefore, the land-based emissions for those countries that are not included in Europe, but are part of the rectangular area of the domain have been added to the dataset. These "missing countries" include parts of North Africa, the Middle East and the Eastern European, Caucasus and Central Asia (EECCA) countries. Emissions for these regions were taken directly from the EDGAR v4.3.2 inventory (Crippa et al., 475 2018) for air pollutants (covering 1970-2012). From 2013 onwards emissions for these regions have been kept equal to 2012 levels. EDGAR source categories were converted to GNFR categories, and since the resolution in the EDGAR inventory (0.1° x 0.1°) is a factor 2 coarser compared to the resolution in this inventory, each EDGAR grid cell was divided in half, where both halves were each assigned 50% of the emission from the original EDGAR grid cell.


## 2.7    Speciation profiles, temporal profiles and emission height

For the pollutants which are essentially groups of pollutants (PM, NMVOC), a speciation into actual components has been calculated and provided along with this inventory. At the most detailed sector level, the PM and NMVOC were split in various components for each source. PM emissions are divided into EC (elemental 485 carbon), OC (expressed in full molecular mass), sulphate (SO4), sodium and other minerals. NMVOC emissions have been split in 23 different hydrocarbon groups. Eventually, PM and NMVOC profiles are provided per country and per year, reflecting the different shares of subsectors and fuels in each situation. In the case of PM, the profiles distinguish between fine (<2.5μm) and coarse mode (2.5-10μm) particles.

Temporal emission profiles can be used to break down annual emissions into hourly values by means of applying 490 factors representing the month, the day in the week and the hour in the day. These are default profiles per GNFR sector, to be applied for all pollutants, countries and years, largely based on the earlier temporal profiles provided with the TNO_MACC-II inventory (Kuenen et al., 2014). Recently a more detailed set of temporal profiles (CAMS-TEMPO) has been proposed (Guevara et al., 2021), for which the evaluation is currently ongoing. Based on the results of this evaluation, the default temporal profiles may be updated in the future.



Finally, for the emission height a default height profile per sector is included, which accounts for the average effective emission height (including plume rise), based on earlier work (Bieser et al., 2011). Especially in sectors which include stacks, emissions are released into the atmosphere at higher altitude, which has important consequences for air quality especially close to these sources.

The PM and NMVOC speciation files, the default temporal emission profiles as well as the default emission
height profiles are all available as separate files which are provided along with the gridded data files.

# 3 Results

## 3.1 Resulting emissions

Table 2 shows the total emissions for each pollutant for selected years, as well as the emission trend. It shows that
for each air pollutant the emissions have decreased during the whole period. However the level of reduction for the period 2000-2017 as a whole differs significantly between pollutants, with the largest reductions for $SO_2$ emissions and lowest reductions for $NH_3$. In Table 2 the trend is calculated separately for the period 2000-2010 and for 2010-2017, which shows that for most pollutants the reductions in the 2000s have been significantly larger than in the 2010s. For $NO_x$ and PM however, reductions are more stable, whereas for $NH_3$ a small increase
in emissions is found between 2010 and 2017.

**Table 2 Emissions for selected years for all pollutants (sum of all countries and sectors, given in kton) and the trend in emissions between 2000 and 2010 and between 2010 and 2017**

|  | 2000 | 2005 | 2010 | 2015 | 2017 | Trend 2000-2010 | Trend 2010-2017 |
|---|---|---|---|---|---|---|---|
| $CH_4$ | 47 425 | 45 147 | 41 431 | 39 665 | 39 448 | -13% | -5% |
| CO | 58 235 | 49 792 | 43 261 | 35 885 | 35 299 | -26% | -18% |
| $NH_3$ | 5 877 | 5 490 | 5 261 | 5 339 | 5 410 | -10% | 3% |
| NMVOC | 15 805 | 13 532 | 11 377 | 9 867 | 9 757 | -28% | -14% |
| $NO_x$ | 17 190 | 16 212 | 12 987 | 11 090 | 10 397 | -24% | -20% |
| $PM_{10}$ | 5 465 | 5 281 | 4 854 | 4 483 | 4 436 | -11% | -9% |
| $PM_{2.5}$ | 3 705 | 3 560 | 3 347 | 3 044 | 3 025 | -10% | -10% |
| $SO_2$ | 16 110 | 12 665 | 8 748 | 7 482 | 6 230 | -46% | -29% |

The emission reductions are not equally distributed over Europe. Figure 3 shows the change in emissions from
2000 to 2017 per country group. The country groups distinguish 4 different regions covering the EU+ countries,





and separately non-EU countries and sea (international shipping). The exact definition of the country groups is provided in the SI (Table S6). It shows a mixed picture between different pollutants. Overall, largest emission reductions were achieved in the western, central and southern EU countries for most pollutants. Smaller reductions of emissions are seen for the non-EU countries and for shipping.


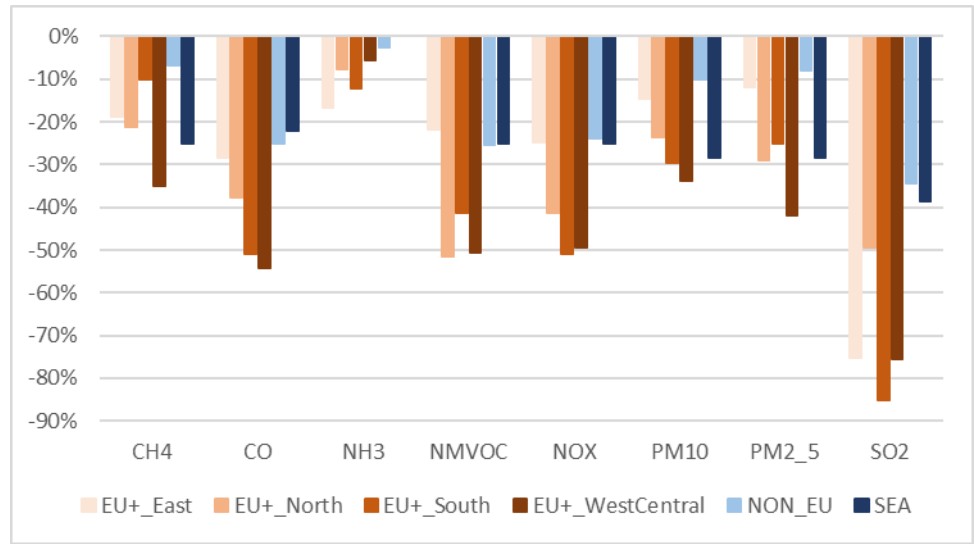

**Figure 3 Emission change between 2000 and 2017 per pollutant and per country group**

Figure 4 shows spatially distributed emissions for 2 selected pollutants (NMVOC and $NO_x$), for the sum of all sectors. In this plot, the resolution is aggregated here by a factor 2 to increase visibility of point sources in the

map. Apart from these point sources (clearly visible as red dots in the maps), other major sources (shipping, road transport) as well as urban areas are shown with higher emissions. For NMVOC, emissions are more diffuse, with small combustion and solvent use as important contributors. But here also point sources are a significant contributor as shown in the map.



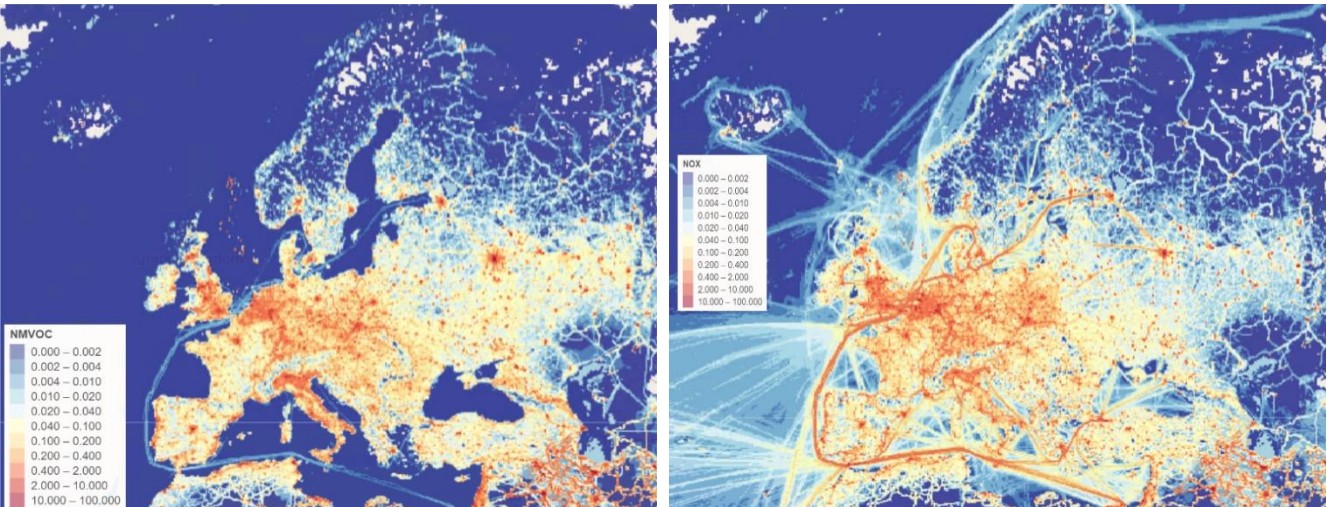

**Figure 4 Spatially distributed emissions of NMVOC (left panel) and NOₓ (right panel) for 2017, in both cases the total for all sectors is shown**

The difference between spatially distributed emissions in 2000 and 2017 is shown in Figure 5 for $PM_{2.5}$ emissions from small combustion (GNFR C) and $NO_x$ emissions from road transport – diesel exhaust (GNFR F2). Both examples show reductions in some countries and increases in other countries, but also within countries. The latter is due to differences in the relative contribution of underlying specific emission sources which are spatially distributed using different parameters. Figure 5 also indirectly illustrates that the EU has more coordinated policies on road transport engine technology and associated emissions than for residential combustion which shows a much more variable development. The reason why road transport $NO_x$ emission in eastern Europe do not follow the trend of the other EU regions is that the growth of road transport activity is larger than the emission reduction per vehicle.

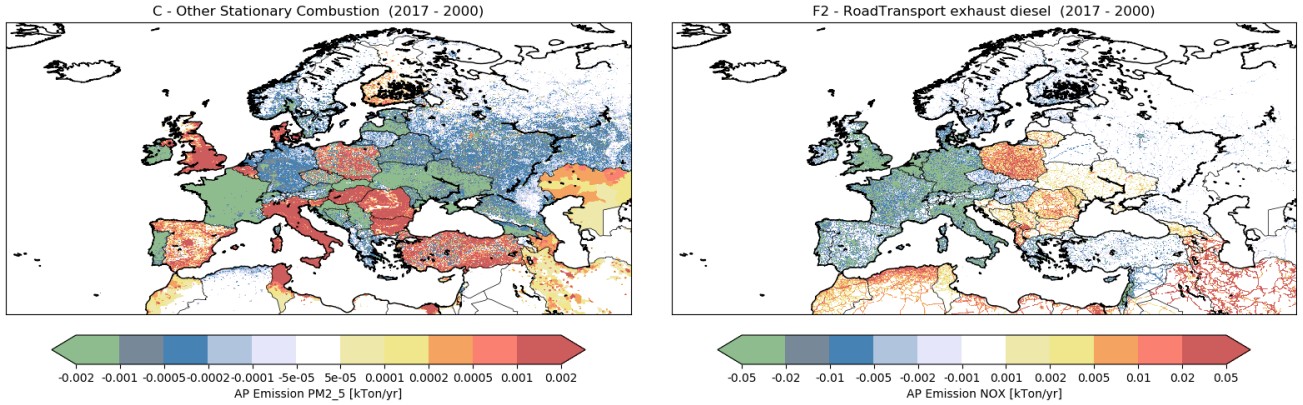

**Figure 5 Difference between emissions in 2017 and 2000 for $PM_{2.5}$ from small combustion (left panel) and $NO_x$ from road transport, diesel exhaust (right panel) (figures are excluding international shipping)**



### 3.1.1 Shipping

Shipping emissions include both shipping at sea and on rivers. For the main air pollutants, the contribution of inland shipping at the European scale is limited, typically 3-4% for PM and $NO_x$ and <0.5% for $SO_2$. Nevertheless, in major rivers and near large harbours, inland shipping emissions may be important for local air quality. As an example of the output from the STEAM model, Figure 6 shows the distribution of shipping emissions in the North Sea and the Benelux. This illustrates major shipping routes at sea, high emissions in and

close to the major ports (Rotterdam, Amsterdam, Antwerp). Further transport along the main rivers is shown in the inland shipping map (center) which shows higher emissions along the inland waterways in the Netherlands, Belgium and Germany.

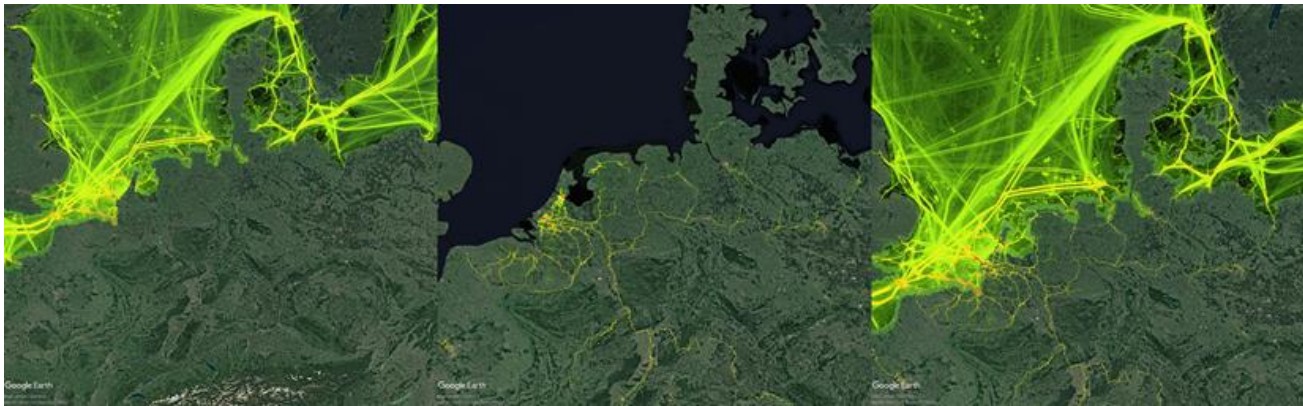

**Figure 6 Example emission distribution of CO₂ for sea shipping (left), inland shipping (middle) and sum of both (right) for a**
**specific region covering parts of the North Sea, Baltic Sea as well as inland waterways (example for 2016, at high resolution of 1/60° x 1/120°) (figures using © Google Maps)**

As explained in Sect. 2.4, gridded emissions from the STEAM model were only available for 2013 onwards. For earlier years, scaling factors were developed for the shipping emissions to estimate emissions in the year 2000-2012 by sea, taking into account environmental control measures such as the Sulphur Emission Control Areas

(SECA). This is illustrated in Figure 7 where implementation of SECA on the North Sea in 2010, and more stringent in 2015, is clearly visible in the trend of $SO_2$ emissions.



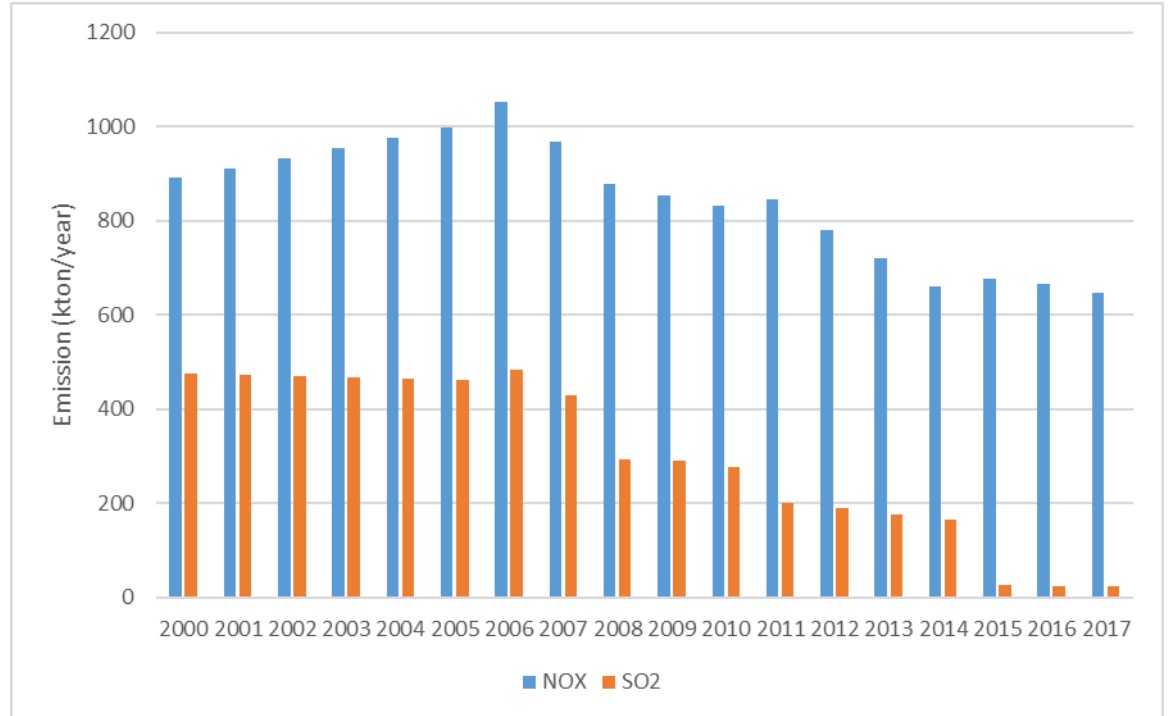

**Figure 7 Emissions of NOₓ and SO₂ for the North Sea (NOS) including the English Channel (ENC) over 2000-2017.**


## 3.2  Comparison to other inventories

### 3.2.1  Comparison to earlier versions

CAMS-REG-v4.2 does not only add new years to the inventory compared to earlier CAMS-REG versions, it also provides updated emissions for the entire time series back to 2000. To illustrate the difference with earlier
versions, Table 3 shows the relative change between total emissions for the sum of all EU+ countries for the same year with the two most widely used predecessors. TNO_MACC-III is an extension of the TNO_MACC-II database (Kuenen et al., 2014), covering the years 2000-2011. CAMS-REG-v2 is an earlier version of the current CAMS-REG-v4 dataset, covering the years 2000-2015. Differences between the same years in the different inventories are partly related to methodological changes in the CAMS-REG inventory, but most can be explained
by recalculations of official reported emissions of air pollutants by each country. For TNO_MACC-III, emission data from reporting year 2013 are used as the basis, for CAMS-REG-v2 this concerns emission data as reported in 2017, while this dataset builds on emission data as reported in 2019.  The Table shows that PM and NH₃ emissions are typically higher in CAMS-REG-v4.2 compared to TNO_MACC-III, while NOₓ, SO₂ and CO emissions are typically lower.

One of the significant changes between this dataset and its predecessors is the approach to AWB as discussed in Sect. 2.5. Table 3 shows that CAMS-REG-v4.2 for incomplete combustion related species like CO and PM₂.₅ is 6



% lower that its immediate predecessor CAMS-REG-v2.2.1. While this is the net sum of various sources being adjusted downward and upward, an important contribution comes from the revised AWB estimate based on earth observation data. Over the entire European domain AWB now contributes 3.1% and 3.3% to total CO and $PM_{2.5}$

emission respectively. This used to be 8.2% and 11.2%, respectively in CAMS-REG-v2.2.1.

**Table 3 Relative change between CAMS-REG-v4.2 and its predecessors (TNO_MACC-III and CAMS-REG-v2.2.1) for each pollutant for selected years, for the EU+ as a whole**

|  | Difference against TNO_MACC-III | | | Difference against CAMS-REG-v2.2.1 | | |
|---|---|---|---|---|---|---|
|  | **2000** | **2005** | **2011** | **2005** | **2010** | **2015** |
| **$CH_4$** | 6% | 6% | 2% | -1% | -1% | 0% |
| **CO** | -5% | -2% | -9% | 1% | -3% | -6% |
| **$NH_3$** | 9% | 9% | 6% | 0% | -1% | -2% |
| **NMVOC** | -2% | -1% | -5% | 2% | 1% | -1% |
| **$NO_X$** | -4% | -4% | -7% | 0% | 0% | 0% |
| **$PM_{10}$** | 7% | 14% | 12% | 2% | 1% | -1% |
| **$PM_{2.5}$** | 2% | 6% | 7% | -3% | -4% | -6% |
| **$SO_2$** | -5% | -6% | -14% | 1% | -4% | 1% |

Figure 8 shows the trends in emissions for the EU+ countries from the most widely used earlier versions: TNO_MACC-III and CAMS-REG-v2. It illustrates the difference is not static in time and may change from year to year. Most of the differences can be explained by changes in reporting, which are in turn related to improved understanding of emissions and improved guidance for emission estimation.



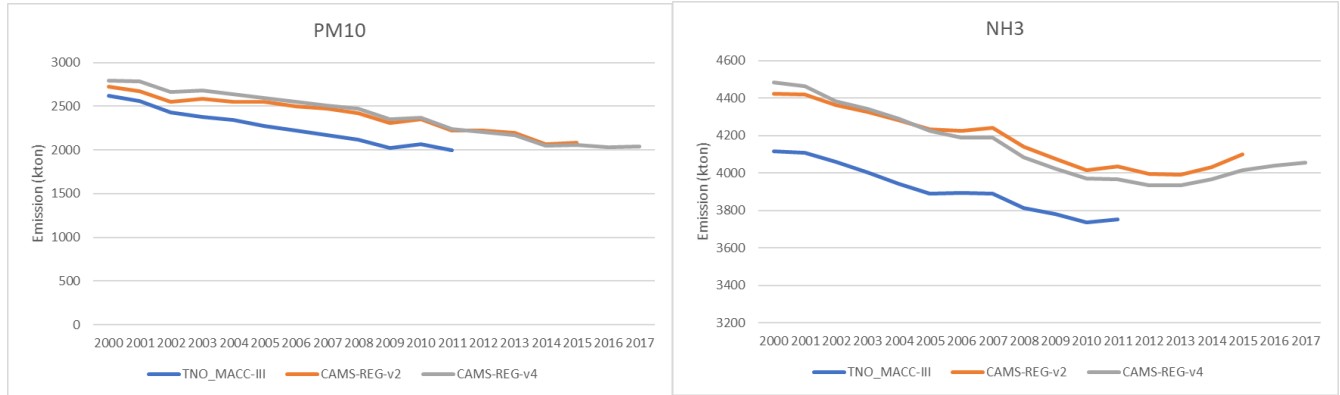


**Figure 8 Comparison between CAMS-REG-v4.2 and its predecessors (TNO_MACC-III and CAMS-REG-v2.2.1) , for EU+ as a whole for PM₁₀ (left panel) and NH₃ (right panel)**

Figure 10 shows a country specific comparison for PM$_{2.5}$ emissions from GNFR C (small combustion) in TNO_MACC-III, CAMS-REG-v2 and CAMS-REG-v4, for the EU+ countries where reported data are used as
the basis, therefore representative for the reporting of PM2.5 emissions from this source category in different years (2013, 2017 and 2019, respectively). The figure shows significant changes for some countries (e.g. EST, LTU, GBR, ESP, ITA, ROU) but only very small changes for others (e.g. DNK, NOR, DEU, FRA, SVK). These differences are significant and related to the inclusion of condensables in the emission inventories (which his further discussed in Sect. 4).

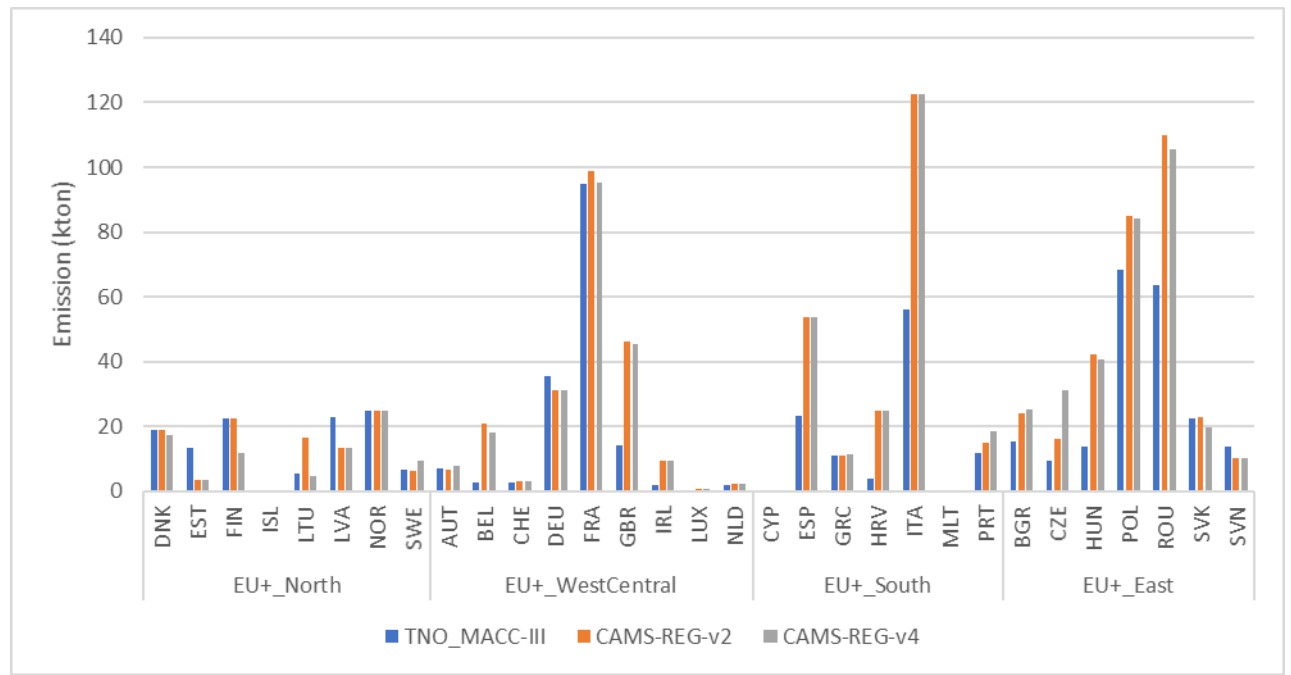


**Figure 9 PM$_{2.5}$ emissions from small combustion (GNFR C) in TNO_MACC-III, CAMS-REG-v2 and CAMS-REG-v4 for the year 2010 and EU+ countries**



### 3.2.2   Comparison to EDGAR

EDGAR (Emissions Database for Global Atmospheric Research) (Crippa et al., 2018, 2020) is a widely used global emission inventory, which uses a bottom-up approach for all sectors based on activity data (energy statistics, industrial production, etc.) combined with emission factors, developed independently from the national inventories from individual countries. Table 4 shows a comparison between the results from this inventory and EDGAR v5.0 (Crippa et al., 2020), for the year 2015. The comparison is made separately for the EU+ countries

(where CAMS-REG is largely based on reported data from the countries) and non-EU countries (where CAMS-REG is largely based on the GAINS emissions). The Russian Federation is excluded from this analysis since EDGAR covers the total of the country, while CAMS-REG only includes the European part of Russia.

Total emissions from EDGAR and CAMS-REG at European scale differ considerably between pollutants, as illustrated in Table 4, especially for non-EU countries. However, also for the EU+ countries significant

differences are seen for $CH_4$, $NH_3$, NMVOC and $SO_2$ in particular. In most cases EDGAR emissions are higher, except for PM in non-EU countries where CAMS-REG-v4.2 provides higher emissions.

| | EU+ countries | | | Non-EU countries | | |
|---|---|---|---|---|---|---|
| | CAMS-REG-v4.2 | EDGAR v5.0 | Difference | CAMS-REG-v4.2 | EDGAR v5.0 | Difference |
| $CH_4$ | 18 763 | 25 495 | 36% | 5 967 | 9 976 | 67% |
| CO | 20 250 | 22 025 | 9% | 7 493 | 8 494 | 13% |
| $NH_3$ | 4 016 | 5 717 | 42% | 892 | 1 798 | 102% |
| NMVOC | 6 188 | 8 372 | 35% | 1 450 | 2 668 | 84% |
| $NO_X$ | 7 218 | 7 676 | 6% | 1 822 | 2 365 | 30% |
| $PM_{10}$ | 2 056 | 2 158 | 5% | 1 332 | 957 | -28% |
| $PM_{2.5}$ | 1 318 | 1 375 | 4% | 934 | 642 | -31% |
| $SO_2$ | 2 792 | 4 649 | 66% | 3 311 | 3 789 | 14% |

**Table 4 Comparison between total annual emissions for 2015 from CAMS-REG-v4.2 and EDGAR-v5.0 for EU+ countries (left) and non-EU countries (right), excluding the Russian Federation (emissions in kton)**

Figure 10 shows the difference between CAMS-REG-v4 and EDGAR-v5.0 on a GNFR sector level, for $NO_X$ and

NMVOC. To make this comparison, EDGAR emissions (which use the IPCC classification) were converted to GNFR sector classification. The match is not always perfect, which is illustrated by the figure for NMVOC, where the EDGAR-v5.0 emissions for GNFR category B (Industry) also include emissions from solvents (GNFR E), which explains the large discrepancy there. Also emissions from GNFR I (Off-road) are partly included in



other sectors, in particular in GNFR C. Key differences are seen for GNFR D (fugitives) and GNFR J (waste)
where EDGAR includes significantly higher emissions, but also for agriculture livestock (GNFR K) there is a
discrepancy since NMVOC emissions from this source are not included in CAMS-REG (see Sect. 2.2.4).

For $NO_x$, the comparison shows a large difference for shipping (GNFR G) where EDGAR emissions are
significantly higher. This may be related to the allocation of shipping emissions between countries and sea
regions, as this comparison excludes international shipping. On the other hand, EDGAR emissions are
significantly lower for GNFR I (off-road and other transportation) which is partly related to the sector allocation
issue. Another difference is seen in GNFR L (other agriculture), which could be related to the inclusion of soil-
$NO_x$ in EDGAR which is not included in CAMS-REG-v4 (see Sect. 2.2.4).

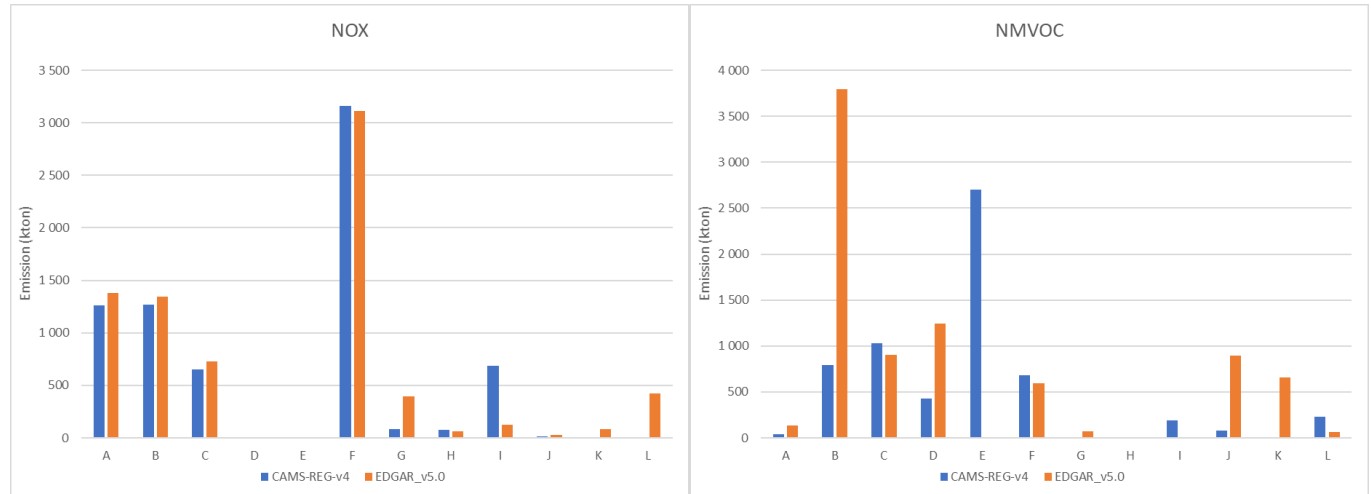

**Figure 10 Comparison between CAMS-REG-v4.2 and EDGAR-v5.0 for $NO_x$ and NMVOC for the year 2015 for EU+ countries**

Figure 11 shows the emission trends between 2000 and 2015 both in CAMS-REG-v4.2 and EDGAR-v5.0 for 5
selected pollutants. It is shown that generally the trends are comparable in both datasets, but the downward trend
in CAMS-REG is stronger for each of these pollutants compared to EDGAR. This difference in trend is most
notable for NMVOC, $SO_2$ and PM.



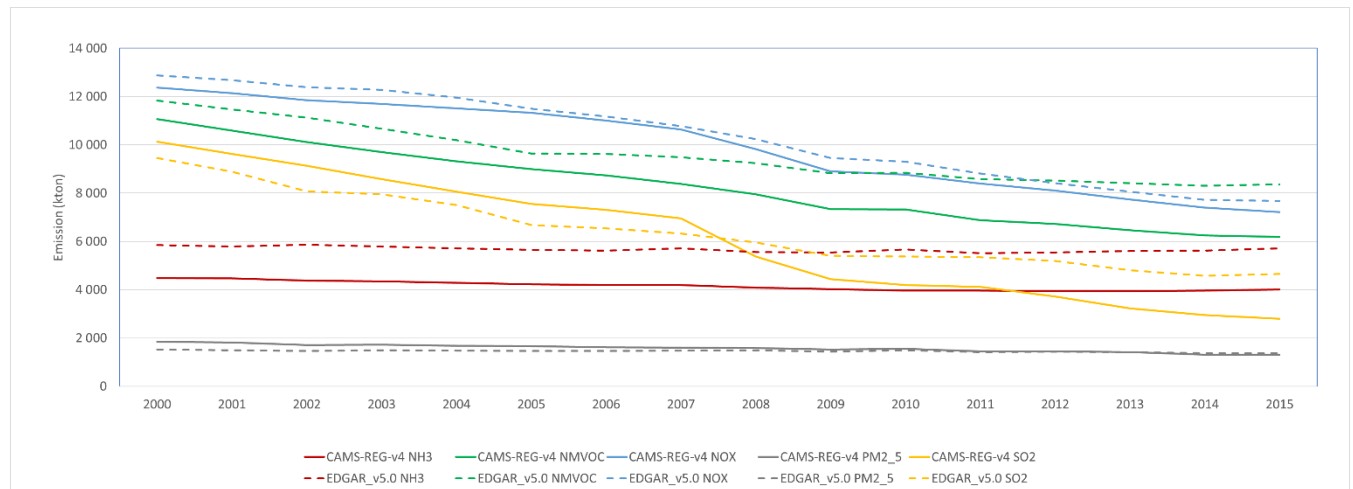


**Figure 11 Trends in CAMS-REG-v4.2 (solid lines) and EDGAR-v5.0 (dashed lines) for 5 key pollutants between 2000-2015 (Russian Federation is excluded from the comparison)**

For a comparison of previous versions of the CAMS-REG and TNO-MACC data to the inventories made by the US-EPA we refer to (Pouliot et al., 2015).

**4 Discussion**

The inventory described in this paper is an updated and improved inventory of the earlier described TNO_MACC inventories (Kuenen et al., 2014). Compared to this older inventory, the main changes are the further increased resolution (0.1°x0.05°, was 0.125°x0.0625°) and the sectoral classification from SNAP to GNFR. The main reason for doing so was to allow easier (inter)comparison with national gridded data reported to EMEP and
international datasets like EDGAR, both at 0.1° x 0.1°. The GNFR sectoral classification is used for official inventory reporting, and harmonization is beneficial for comparison and data exchange. At the same time, the inventory has been fully updated, especially with regard to the spatial distribution. This improves the representation of emissions, especially when looking at larger time scales. For instance, a new population map was introduced, which better represents the actual population density over Europe. By using this population map
for 3 different years (2005, 2010, 2015) also the changes of the population distribution over time can be represented in the emission distribution, since urbanisation plays an important role in some countries over the almost 20 year period. Additionally, the representation of point sources is significantly improved by incorporation of the E-PRTR (and associated datasets) which provide a robust representation of point source emissions over time, taking into account changes such as opening or closure of specific facilities as this may have
significant impact on emissions in specific areas.

The CAMS-REG-v4.2 emission inventory was constructed by combining different available datasets, similar to its previous versions. Wherever possible the choice of which dataset to use and in which situation, is based on objective criteria. For instance the use of reported emission data for EU+ countries but not using reported data for other countries. This assessment is based largely on the experience of working with these datasets in earlier years,



where for the non-EU+ countries the country reported data were not considered to be fit for purpose (Denier Van Der Gon et al., 2010; Kuenen et al., 2014). It was also found that even for the countries where reported emissions are expected to be good quality, errors and inconsistencies may exist. Key examples include NMVOC from agricultural husbandry and $NO_x$ from all agriculture, which were excluded both, for the latter also to avoid double counting since there are air quality models that calculate $NO_x$ from agricultural soils themselves. A thorough

assessment of the reported data for each EU+ country was performed. Since mistakes or inconsistencies in country reporting of emissions may concern various aspects (e.g. missing years, missing sectors, unrealistic distribution of emissions over sectors, etc.), it is difficult to apply a set of fixed rules to filter out errors, as these may miss specific errors as well as trigger false positives. For some further examples on inconsistency in reporting we refer to Kuenen et al. (2014). This makes that the use of expert judgement to make choices on what

(not) to use is a necessity.

The combination of different datasets and frequent use of expert judgement make the assessment of uncertainties more difficult. In the national emission inventories provided by individual countries, uncertainty assessment is one of the elements to be taken into consideration. The main goal of assessing uncertainties in national emission inventories is to help prioritize inventory improvements at national level by improving first those sectors with

relatively high uncertainty, thus efficiently reducing the uncertainty of the inventory as a whole. The EMEP/EEA Guidebook also includes a description of the methodology to be followed for such an uncertainty assessment (EEA, 2019a), which also provides indicate uncertainty ranges for activity data and emission factors depending on the source of the data. The EMEP/EEA Guidebook also provides default activity ranges, based on the assumption that all sources are calculated using activity data and emission factors. Direct measurements of

emissions in individual large installations would reduce uncertainty, therefore these values could be seen as an upper limit. Table 5 provides these default ranges. It should be noted that these do not take into account the uncertainty in spatial distribution of emissions.

|  | SO₂ | NOₓ | NMVOC | CO | NH₃ | PM |
|---|---|---|---|---|---|---|
| **A_PublicPower** | 10-30% | 20-60% | 50-200% | 20-60% | Order of magnitude | 50-200% |
| **B_Industry** | 10-60% | 20-200% | 20-200% | 20-200% | Order of magnitude | 50-200% |
| **C_OtherStationaryComb** | 10-30% | 50-200% | 50-200% | 50-200% | Order of magnitude | 100-300% |
| **D_Fugitives** | 50-200% | 50-200% | 50-200% | 50-200% |  | 100-300% |
| **E_Solvents** |  |  | 20-60% |  |  |  |
| **F_RoadTransport** | 10-30% | 50-200% | 50-200% | 50-200% | Order of magnitude | 50-200% |



| | | | | | | |
|---|---|---|---|---|---|---|
| **G_Shipping, H_Aviation, I_Offroad** | 20-60% | 100-300% | 100-300% | 100-300% | Order of magnitude | 100-300% |
| **J_Waste** | 20-60% | 20-60% | 20-60% | 50-200% | Order of magnitude | 50-200% |
| **K_AgriLivestock, L_AgriOther** | | 100-300% | 100-300% | 100-300% | 100-300% | Order of magnitude |

**Table 5 GNFR categories and their estimated uncertainty based on the default approach to emission inventories using activity data and emission factors, and the typical source of the data. Adapted from (EEA, 2019a).**

However, despite requirements to do so, not all countries perform such an uncertainty assessment, which is also concluded in a recent report (Schindlbacher et al., 2021). The uncertainty values reported by different countries are shown in Figure 12. This illustrates a wide range of reported uncertainties in total emissions, ranging between <10% and >50% for $NO_x$ and between 5% and nearly 40% for $SO_2$. The countries not shown here did not report quantitative information on uncertainties.

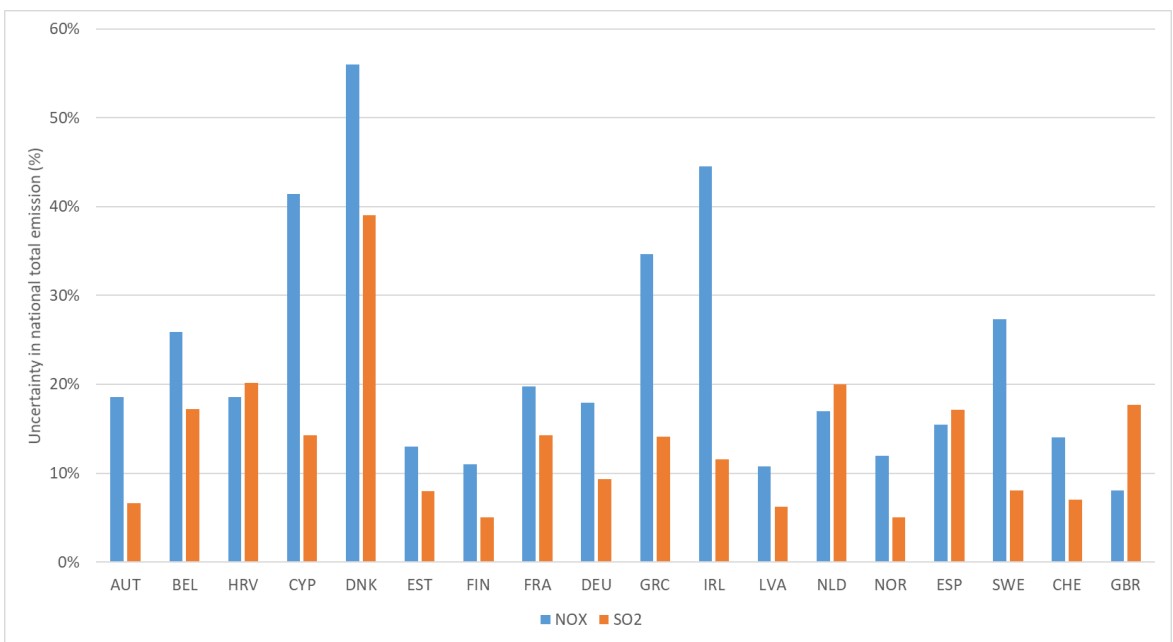

**Figure 12 Uncertainties in total emissions of $NO_x$ and $SO_2$ as reported in the Informative Inventory Reports submitted by the countries in 2020 (data taken from (Schindlbacher et al., 2021))**

The uncertainties shown in Figure 12 are generally lower than those reported in Table 5, since the emission inventories for these countries are well established, in many cases using directly measured emission data for important sources, which reduces the overall uncertainty. A large variation is shown between uncertainties for different countries, even for the same pollutant. This relates to the methodology and level of detail in which these





uncertainty assessments are done and reported on differ per country. Also, the estimation of uncertainties for
individual parameters such as activity data or emission factors often requires a significant amount of expert
judgement in the absence of real data on uncertainties. Since countries perform these uncertainty assessment
individually, resulting overall uncertainty estimates differ significantly per country, and applying the available
uncertainty estimates directly to a European scale inventory will introduce inconsistencies in these uncertainties
between countries. This poses significant shortcomings in the direct uptake of uncertainty data from country
emission inventories in the European-wide uncertainty assessment.

An alternative to using country data on uncertainties would be to perform a complete uncertainty assessment.
This requires uncertainties to be estimated for all parameters involved in the emission estimation, including
activity data, emission factors, and something that accounts for the uncertainty in spatial distribution. A first
attempt to quantify uncertainties in emissions in such a way was recently made by Super et al. (2020) for $CO_2$ and
CO. Whereas the annual country-level $CO_2$ emissions have a relatively low uncertainty, for CO the emission
factors had the largest contribution to the uncertainties. Also the uncertainty in the spatial proxies was assessed,
showing a significant contribution to the uncertainty in the gridded inventory, especially at higher resolution.
Another example of estimating uncertainties for greenhouse gases at global scale was made recently for the
EDGAR inventory (Solazzo et al., 2021), but this study did not consider the uncertainty in the spatial distribution
component. Given the variety of datasets used in CAMS-REG, the focus on air pollutants which have more
uncertain EFs compared to greenhouse gases and the almost 20-year time span, more work is needed to
independently estimate emission inventory uncertainties, including spatial error correlations. This could be
considered a future priority to develop.

Given the difficulty in deriving direct uncertainties of the emissions at grid level, comparisons to other
independent emission datasets are a useful way to identify key differences and apparent uncertainties. In this
paper it is shown that EDGAR-v5.0 emissions differ significantly from CAMS-REG emissions, especially for
non-EU countries (Table 4). Such differences could be regarded as indicative of the uncertainties associated with
present-day anthropogenic emissions but this is a topic that deserves more attention and effort in the future, as
mentioned earlier.

For the comparison to the earlier versions of this dataset it is found that most of the differences in the emission
estimates can be traced back to differences in national emission reporting. European countries are obliged to
report their emissions in the national inventories for each year in the time series annually (back to 1990), and at
the same time every year improvements and updates are made to the inventories, incorporating new information
on activity data or emissions factors. This means that every year the entire time series is revised, which may incur
significant changes to the overall emissions in each country. Since each country has its own inventory team with
its own challenges and data sources, such revisions do not always go in the same direction.

One example of large changes to earlier reported emissions over time can be found in the PM emissions from
small residential combustion. The main reason for adjustments over time is the increasing insight and awareness
of the role that condensable, mostly organic compounds play in total PM emission from this source sector. These
compounds are emitted in gaseous phase, but immediately after leaving the stack or chimney may condense to
form particles. Depending on the measurement device used, these particles may or may not be captured in the PM
emission measurement (Denier van der Gon et al., 2015). In the EMEP/EEA Guidebook (EEA, 2019a), the




reference document for compiling air pollutant inventories in Europe, condensables were consistently introduced for small combustion of biomass as part of the 2016 update, which gradually encouraged more and more
countries to report on this basis. The main motivation for doing so is the better understanding of ambient PM as supported by better prediction skills of air quality models and better agreement with observations (Bergström et al., 2012; Simpson et al., 2020). Figure 9 shows PM$_{2.5}$ emissions reported in this inventory and 2 of its predecessors, where the emission data were based on reported data as reported in 2013, 2017 and 2019, respectively. The differences between the datasets are representative for the differences in reporting from the
inventories, which in turn are to a large extent related to the inclusion (or not) of condensables. Amongst others, it can be observed that between 2013 and 2017 the reported emissions in e.g. Belgium, the United Kingdom, Spain, Italy and Romania increased significantly, which was confirmed to be caused by the inclusion of condensables in these cases.

Given that the CAMS-REG inventories are derived to support air quality assessment at European scale, not all the
details for specific national circumstances may be included. While specific national circumstances are incorporated for as far as sector totals are concerned (through the direct use of country reported data for EU+ countries), the spatial distribution methodology is uniform over Europe whereas individual countries may use detailed distribution proxies specific to their country. This means that when zooming in, differences with these national distributions and also with local bottom-up estimates of emissions are typically found. These are to a
large extent related to the spatial distribution component (Trombetti et al., 2018).

Due to the consistency in which country reported data are being processed the CAMS-REG datasets can also be used to investigate trends and derive information on specific sources. An example is the analysis presented by (Denier van der Gon et al., 2018) on the increasing importance of non-exhaust PM emissions from road transport. This is a separate source category in CAMS-REG-v4.2 (GNFR sector F4). Stringent EU policies over time
succeeded in reducing the road transport exhaust PM emissions and by now "non-exhaust" emissions from brake wear, tyre wear and road abrasion have started to dominate the PM$_{10}$ emission from road transport. As the share of coarse PM is relatively high in wear emission, the PM$_{2.5}$ emission from road transport may still be dominated by exhaust emissions. An important aspect of having a separate category in the emission data for wear emissions is that different chemical composition profiles can be applied, for instance to estimate various heavy metal
emissions from non-exhaust road transport emissions (Denier van der Gon et al., 2018).

The CAMS-REG inventory is widely used in modelling activities worldwide because of its consistent approach and longer time series it is especially useful to support large intercomparison activities between models such as the Air Quality Modelling International Initiative (AQMEII) (Im et al., 2015). Another example where CAMS-REG is used is the HTAP (Hemispheric Transport of Air Pollution) inventory (Janssens-Maenhout et al., 2015).
Here, a global emission inventory (EDGAR) was updated by nesting specific regional inventories in specific regions of the world (e.g. CAMS-REG) to improve our understanding hemispheric transport of air pollution. Finally, with the increasing resolution of emission inventories to support modelling exercises at local to regional scale, the temporal distribution of emissions within the year becomes increasingly important. A specific set of temporal profiles was derived specifically to be applied with the CAMS emission inventories including CAMS-
REG. These profiles take into account more detailed temporal variations compared to the default temporal profiles provided along with this dataset, such as the variation of emissions with meteorology (e.g. temperature dependency of residential heating) (Guevara et al., 2021).



## 5    Data availability

Gridded emission maps with all pollutants are available for each year. The files are provided as NetCDF
(Network Common Data Format) at a resolution of 0.05°x0.1° (latitude-longitude) for the European domain
(30°N-72°N, 30°W-60°E) (Kuenen et al., 2021). The emission data in the grids represent annual data per grid
cell, all emissions are given in kg. Access is provided through the Emissions of atmospheric Compounds of
Ancillary Data (ECCAD) system, which will be complemented with access through the ECMWF Atmosphere
Data Store (ADS) as soon as this is technically feasible.

Since the ECCAD system requires a registration and login, for the purpose of the review process a sample of the
emission files has been made available for download without any login or registration requirement. This sample
includes data for the year 2017 only and also all the supporting documentation including PM and VOC speciation
and temporal and height profiles. The sample dataset is available through https://eccad.aeris-data.fr/essd-surf-
emis-cams-reg/.

## 6    Conclusions and outlook

The current CAMS-REG-v4.2 emission inventory is developed in support of air quality modelling activities at
European scale. It incorporates official national emission estimates from countries to the extent possible to
facilitate the use of this inventory for policy applications, and uses a uniform spatial distribution methodology
across Europe to ensure a comparable and consistent emission grid across Europe. CAMS-REG-v4.2 is the latest
version of the a series of emission inventories that was developed in support of modelling. Since in addition to
the gridded emissions, speciation profiles for PM and NMVOC as well as default information on temporal and
height distribution are provided along with the dataset, it provides an excellent starting point for air quality
modelling at European scale. On the other hand, the use of country reported data also implies that for some
sectors there are limitations when the consistency in reporting is limited. Examples include PM emissions from
residential combustion, but also agricultural NMVOC emissions (which are currently excluded from the CAMS-
REG inventory). These specific sources should be looked at in the future to work towards a consistent
representation of these sources in the CAMS-REG inventory. Also non-reported sources such as PM from
resuspension could be considered. In addition to that, developing uncertainties for the emissions in this dataset is
a difficult task given the combination of different data sources and the limited availability of uncertainty
estimates for these datasets. However, it will be important for the years to come to assess the uncertainties of
modelled concentrations and also to compare emission estimates from inventories to those derived from
observations.

## Supplement

The supplement related to this article is available online at (to be added)

## Author contributions





JK coordinated and processed all the different emission data sources, with support from SD and IS. Methodological decisions and choices were made through discussions with AV and HvdG. The spatial
distribution proxies were mostly prepared by SD with the help of AV, IS and JK. SD and AV developed the point source databases used in this study. HvdG gave feedback on the whole inventory development and steered the directions. JPJ provided the shipping emissions. JK prepared the paper, with specific input sections from SD, AV and HvdG. IS reviewed the paper as a whole prior to submission.

**Competing interests**

The authors declare that they have no conflict of interest.

**Special issue statement**

(to be added)


**Acknowledgements**

The research leading to these results has received funding from:

- the Copernicus Atmosphere Monitoring Service (CAMS), which is implemented by the European Centre for Medium-Range Weather Forecasts (ECMWF) on behalf of the European Commission.
- The European Union's Horizon 2020 research and innovation programme under grant agreement No. 776186 (CHE project, coordinated by ECMWF).
- The European Union's Horizon 2020 research and innovation programme under grant agreement No. 776810 (VERIFY project, coordinated by CEA/LSCE).

The authors would like to give special thanks to the GFAS team and Johannes Kaiser for providing the spatially
distributed emissions from agricultural waste burning, and the discussion regarding their use and uptake in emission inventories.

The authors would like to express their thanks to CEIP for making available reported data of air pollutant emissions by all European countries, and to IIASA for making available GAINS emissions through its online tool. The EU Joint Research Centre (in particular Adrian Leip) is thanked for providing spatially distributed
proxy maps that are used for agricultural emissions.

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
