# Peer review of "CAMS-REG-v4: a state-of-the-art high-resolution European emission inventory for air quality modelling"

_Earth System Science Data, 2021_

## Author Response (AR1)

Reviewer 1

(A = Author, R = Reviewer)

R: The manuscript describes an emission inventory developed for the European domain for a 18-year time series (2000–2017) at high spatial resolution, designed to support air quality modelling. It reports emission of $NO_x$, $SO_2$, NMVOC, $NH_3$, CO, $PM_{10}$ and $PM_{2.5}$ and $CH_4$. This database is an updated and improved inventory (TNO_MACC inventories). It is complemented by other national/international databases. Use official reported emission data from national inventories for both the greenhouse gases and the air pollutants and redistribute it spatially according to some proxy information. Also, for regions with poor data uses estimations from IIASA GAINS model. This paper describes the methodology used to derive the CAMS-REG inventory, version 4, covering years 2000-2017. Finally, it compares to EDGARv5.0 international inventory and early versions of the same CAMs inventory group.

The manuscript is well described and clearly documented. It is easy to read and follow. It includes a good effort to harmonize different sources of information (specially from eastern/southeastern countries) to conform an adequate state of the art inventory useful for air quality modelling and climate change. Worth mentioning is the description and impact of national inventories uncertainties. So, I encourage its publication in the present issue.

Minor comments.

A: The authors want to thank the reviewer for the positive feedback and constructive comments. Below we provide a point-by-point answer on each of the minor comments raised.

R: Point sources. Reading your manuscript, I understand, that the EPRTR databases includes emissions from both fuel consumption and processing. You only had to organize and classified the information (by fuel, industry type, and so on). But you have not calculated the emissions using activity data + emissions factors. Eventually any calculation was provided by IIASA-GAINS model. Is this correct?. So, the main job was to harmonize the time series and eventually correct some missing/mistaken data. If such a complete database is available, why is there important differences with your previous version of the inventory or EDGAR (Figure 10, or Figure 8, although I understand that this figure is for total emissions).

A: The reviewer is correct in the assumption that the emission calculation itself (totals by country, sector, pollutant, year) are not dependent on E-PRTR. E-PRTR (and other point source databases) are only used for spatially disaggregating the emissions. For the comparison to our earlier versions the differences are mainly caused by updated reporting (annually by national inventories and to a lesser extent also corrections in E-PRTR for historical years).

EDGAR does not use the E-PRTR database (to our knowledge) in its methodology. Point source emissions are estimated but using different databases (e.g. CARMA for power plants).

Figure 10 shows a large difference in NMVOC emissions for category B, but this is largely the share of category E which is included here (as described in the manuscript just above Fig 10). The other differences are largely from non-point sources and from countries outside the EU (where reported data are either generally of lower quality or not available).

R: Regarding the road transport. Road network are available form Openstreetmap.org . You also say that road traffic is also available for Europe from OTM. Given that information may not have the same quality for all countries and region. What kind of data quality checking have you performed on traffic volume? Have you performed some fuel-mass balances?, Car registry? Tonn/passenger km travelled?

A: Very good remark. We have made comparisons of the traffic volume (total vkms) calculated from these per road type and vehicle category with the vkms provided by the COPERT vehicle emission model. We noted in particular that at highways the total vkms were higher and in urban areas lower, pointing towards an incomplete coverage at smaller roads.

In part based on these comparisons, we have added an additional split in the emissions (which were originally provided per country and main vehicle type) between the road types (urban, rural, highway) based on data from COPERT (last bullet point in Sect. 2.2.4. The distribution maps are applied at this level, which ensures a certain consistency in the total emissions per country, vehicle type and road type. However, within those categories the split to individual roads and the traffic volume on each one of them has not been checked in detail, only visual checks have been made on the maps (e.g. are the highways with highest intensity where they are expected).

We added an additional paragraph to the SI (section on road transport) to describe this.

R: Regarding the shipping sector. Have you directly adopted the STEAM outputs, or was it processed again?. Are STEAM data public available?. Since STEAM is a Model, it has its own uncertainties and proxies to fill their own gaps. Have you performed any kind of double checking the information from this model?. Fuel checking, ports arrivals, tons and passenger movements by ports and so on?.

Outputs of STEAM can be accessed, but the model code, the activity and fleet technical data are not publicly available because contracts with third parties prohibit sharing of commercial data. There are recent comparisons of STEAM predictions

and the EU fuel reporting scheme (EU MRV). One such example can be found in HELCOM Maritime21/4-2.INF (https://portal.helcom.fi/meetings/MARITIME%2021-2021-939/MeetingDocuments/4-2%20Emissions%20from%20Baltic%20Sea%20shipping%20in%202006%20-%202020.pdf) document where vessel level fuel consumption from STEAM was compared with reported totals. In short, average error for vessel level consumption was 20%, whereas the inventory total for 1604 vessels was off by 7.8%. Uncertainties may arise from multiple reasons, but largest deviations are usually observed in cases where vessel technical description is incomplete. Differences arising from different ship modeling approaches were recently reported (Schwarzkopf et al, Atmospheric Environment: X, Volume 12, December 2021, 100132) and the scatter in emission factor assignment in Grigoriadis et al., Atmospheric Environment: X, Volume 12, December 2021, 100142

R: Figure 10: caption should include the sector names for A, B., C…. (or " see Table 5")
A: We added a reference to Table 5 in the caption.

**Reviewer 2**

(A = Author, R = Reviewer)

R: The paper describes an air pollutant emission inventory for Europe which is now widely used in the atmospheric modelling community. The paper is interesting, complete and well written. It will for sure serve as a very welcome reference paper. I therefore recommend the publication in ESSD provided that the following minor comments are considered by the authors.

A: The authors first want to thank the reviewer for the positive feedback and constructive comments. Below we provide a point-by-point answer for each of the minor comments.

R: General: in some earlier documents, CAMS regional air pollutant emissions were referred to CAMS-REG-AP, as opposed to CAMS-REG-GHG for green house gases. Is it because CH4 is included here that the new reference is CAMS-REG, and is there still a CAMS-REG-GHG where CO2 (and other?) emissions would be reported?

A: What is named CAMS-REG in this paper is essentially CAMS-REG-AP. CH4 is (and has been) a part of the AP inventory given its impact on ozone formation. Indeed CO2 is reported as part of CAMS-REG-GHG. We added some explanation in the introduction to explain this.

R: Abstract P1 L15: EU countries are reporting simultaneously to LRTAP and European Commission for the NEC Directive, the second should also be mentioned here.

A: Updated accordingly.

R: Introduction P2L44: Add that this is mainly for the "*European* air pollution community"

A: Updated accordingly.

R: Introduction: It seems that UNFCCC is not just introduced as an analogy but also because it is the reference for CH4 emissions. In that case it would be worth discussing here information about the gridding of emission data reported to UNFCCC.

A: We added 2 sentences to the end of the paragraph that for reporting of emissions under UNFCCC spatially distributed emissions are not considered, and hence no country reported grids are available for CH4.

R: Section 2 P3L89: in the LRTAP process, CEIP also gap-fills nationally reported emissions to produce what they deliver on their website as "emissions as used in

models". Are those used in the methodology? If not a few words are needed on the difference in gap filling methodologies compared to the approach developed here.

A: Good point. The CEIP process is separate from this work, however it does use the spatial distribution component from the CAMS inventory in part of it. We added some text on this to the introduction (where reported gridded data are discussed) including an additional reference to the CEIP gapfilling and gridding report.

R: Section 2.2 P8L206: Unlike soil NOx, NMVOV from animal husbandry and manure application is not included in models biogenic emissions modules. Why GAINS has not been used for gap filling instead of just excluding those emissions?

A: Also GAINS unfortunately does not contain animal husbandry and manure application as sources of NMVOC (at least not the scenario used for this work). We updated the manuscript mentioning this. However we consider this a priority to improve for future versions.

R: Section 2.3.1: P10L262 why is CO2 mentioned here?

A: We added a line in the manuscript explaining that CO2 is added to complete the point source database, and in particular CO2 is used for the gapfilling process since it is reported more completely and consistently than other substances.

R: Section 2.3.1 and 2.3.2: It appears (P11L285) that E-PRTR is not only used as proxy, but also withdrawn from the sectoral GNFR emission. This information is important and somewhat "hidden" in this section on spatial proxy. Please consider including it elsewhere. A word of explanation on the matching between E-PRTR subsectors and GNFR would also be helpful.

A: We added a few lines in Section 2.3 where the generic description of the spatial distribution is discussed to make this more visible.

R: Section 2.3.4: and 2.3.7 P14L416: more details on the proxies for residential emissions would be appreciated. The exact relationship applied to population density and wood proximity should be used as residential emissions are not directly proportional to population density. But it should also be commented whether this only applies to wood combustion. More generally, fuel use for residential emissions are also very different between dense urban centres and suburban areas.

A: The details are already given in the SI (Table S5) which give for residential emissions per subsector and per fuel the proxy used. The definition of total, urban and rural population is already provided in Sect. 2.3.4. We added a line in the discussion that the use of population density for residential is a simplification and that, when zooming in, different cities may behave differently. For wood

consumption, we added some text to Sect. 2.3.7 describing in detail how the distribution map has been derived.

R: Section 2.3.6: as for residential emission, the exact relationship between traffic and emissions should be provided as the reference to "proxies" remains somewhat vague. Is only traffic density (and not speed) taken into account?

A: We assume the reviewer is referring here to road transport emissions instead of residential emissions. Only the total vehicle kilometers per vehicle category and road type are considered as pointed out in Sect. 2.3.6. Speed, traffic jams or any other parameters are not considered, as this would basically require a traffic model to the generate distribution maps which would be large additional effort. We made the additional split between urban, rural and highway in the emission dataset before gridding to take into account the fact that per vkm emissions are typically different in urban and rural areas, and on highways. This way, we take into account a country specific differentiation between the road classes, however when zooming into individual cities or urban areas the approach may be too simplistic. However, since the goal of this inventory is to support European scale air quality modelling work, we feel the approach is justified.

Some lines were added in the text reflecting the above, and also in the first paragraph of the discussion we added some more text on the limitations of the spatial distribution approach.

R: Section 2.7: more references are needed regarding the source of information for NMVOC and PM splits.

A: We have expanded the text in Sect. 2.7 and added multiple references for both the PM and VOC speciation.

R: Section 3.1: P18L525: red dots at large point source locations are not visible in my printout.

A: The resolution of the figures is sufficient to see the red dots in the maps (especially in population where the area sources are less important), but when the 2 maps are displayed next to each other they may be too small. We increased the size of the maps in the submitted document by placing one below the other.

R: Section 3.1: P19L537 could it be that the trend in residential emissions is also affected by inconsistent reporting of condensable in time? This would challenge drawing conclusions on the European coordination of actions to mitigate emissions.

A: Such a time series inconsistency within the time series of one reporting year would create a large jump in the time series which we would have picked up with

our data checks. Also in country reporting time series consistency is a very important aspect where focus on, so we don't expect this to occur.

R: Section 3.2.1 P21L575: suggest replacing "this" by "CAMS-REG-v4.2"
A: Updated accordingly.